# Fulminant and Slowly Progressive Type 1 Diabetes Associated with Pregnancy

**DOI:** 10.3390/ijms26136499

**Published:** 2025-07-06

**Authors:** Eiji Kawasaki

**Affiliations:** Diabetes, Thyroid and Endocrine Center, Shin-Koga Hospital, 120 Tenjin-cho, Kurume 830-8577, Japan; e-kawasaki@tenjinkai.or.jp; Tel.: +81-942-38-2222

**Keywords:** anti-islet autoantibodies, fulminant type 1 diabetes, gestational diabetes, latent autoimmune diabetes in adults, prediction, pregnancy, slowly progressive type 1 diabetes

## Abstract

Type 1 diabetes is classified into three clinical subtypes: fulminant type 1 diabetes, acute-onset type 1 diabetes, and slowly progressive type 1 diabetes, also known as latent autoimmune diabetes in adults. Among these, the fulminant and slowly progressive forms may develop in association with pregnancy and are herein collectively referred to as “pregnancy-associated type 1 diabetes”. Fulminant type 1 diabetes can manifest suddenly during pregnancy, often accompanied by ketoacidosis, posing a significant risk to both the mother and the fetus. Early diagnosis and treatment are, therefore, critical. In pregnant women with no prior history of diabetes who present with marked hyperglycemia (≥288 mg/dL) but relatively low HbA1c levels (<8.7%), fulminant type 1 diabetes should be suspected, and insulin therapy should be initiated immediately. Conversely, women diagnosed with gestational diabetes who test positive for anti-islet autoantibodies are at high risk of developing slowly progressive type 1 diabetes postpartum. For these patients, regular monitoring of blood glucose levels, HbA1c, and endogenous insulin secretion is essential for early detection and management.

## 1. Introduction

Pregnancy-related glycemic dysregulation is typically classified into three categories: (1) pregestational diabetes, in which a patient with pre-existing diabetes becomes pregnant; (2) overt diabetes in pregnancy, where previously undiagnosed diabetes is first identified during pregnancy; and (3) gestational diabetes mellitus, characterized by glycemic dysregulation having developed and first been detected during pregnancy but not meeting the diagnostic criteria for overt diabetes [1,2].

It is well established that pregnancy induces significant immunological changes, which may contribute to the onset of type 1 diabetes during pregnancy or in the postpartum period [3,4]. Clinically, type 1 diabetes is classified into three subtypes based on the mode of onset: fulminant type 1 diabetes, acute-onset type 1 diabetes, and slowly progressive type 1 diabetes (SPIDDM), also known as latent autoimmune diabetes in adults (LADA) [5,6,7]. Among these, fulminant type 1 diabetes and SPIDDM/LADA are the subtypes more closely associated with pregnancy than the acute-onset type 1 diabetes subtype, and are herein collectively referred to as “pregnancy-associated type 1 diabetes”. This review aims to provide an overview of the characteristics, clinical presentation, risk factors, and predictive markers of these two subtypes in the context of pregnancy.

## 2. Literature Search Strategy

Literature searches were conducted in the database of PubMed Central and the Japan Medical Abstracts Society (Ichushi-Web) with keywords such as “pregnancy”, “gestational diabetes”, “diagnostic criteria”, “immune tolerance”, “regulatory T cells”, “genetics”, “type 1 diabetes”, “fulminant type 1 diabetes”, “slowly progressive type 1 diabetes”, “latent autoimmune diabetes in adults”, “GAD autoantibodies”, and “prediction”. Only peer-reviewed articles published in academic journals, focusing on studies that explored the epidemiology, pathophysiology, and clinical implications of type 1 diabetes and pregnancy, written in English or Japanese were selected. Proceedings, letters to the editor, commentaries, animal studies, and studies not reporting data of interest were excluded. The search was last updated on 10 June 2025.

## 3. Classification of Impaired Glucose Tolerance in Pregnancy

### 3.1. Pregestational Diabetes

Diabetes mellitus is a metabolic disorder that causes hyperglycemia due to insufficient insulin action. Prolonged metabolic abnormalities can lead to diabetic complications, causing dysfunction in multiple organs, including the eyes (retinopathy), kidneys (nephropathy), nerves (neuropathy), and blood vessels (atherosclerosis). Insufficient insulin action may stem from reduced insulin secretion (absolute or relative) or insulin resistance (decreased insulin sensitivity). Diabetes is generally classified into two major types: type 1 diabetes and type 2 diabetes. Type 1 diabetes is caused by an autoimmune process that destroys pancreatic islet β cells, resulting in a marked deficiency of insulin. Type 2 diabetes is caused by both insulin resistance and insufficient insulin secretion and is influenced by lifestyle factors such as overeating, physical inactivity, obesity, stress, and aging.

In pregnancy, increased physiological insulin resistance and accelerated starvation during fasting can predispose women to euglycemic diabetic ketoacidosis, even in the absence of marked hyperglycemia (blood glucose < 200 mg/dL). Insulin resistance is also a significant risk factor associated with obesity and overnutrition, which disrupts normal metabolic function in key insulin-responsive tissues, including the liver, skeletal muscle, and adipose tissue. These disturbances manifest as hyperglycemia, hypertension, dyslipidemia, fatty liver, and muscle proteolysis.

Pregestational diabetes refers to two conditions: (1) a diagnosis of diabetes prior to pregnancy and (2) the presence of established diabetic complications, such as diabetic retinopathy. During the critical period of organogenesis, between the fourth and seventh weeks of gestation, maternal hyperglycemia can result in congenital anomalies and malformations. The risk increases with higher HbA1c levels, rising linearly when HbA1c exceeds 7%, with a two- to four-fold increase compared to the general rate of 3–4%. Additionally, the incidence of spontaneous abortion in women with diabetes increases to approximately 30%, roughly twice that of the general population. Dysglycemia in the second half of pregnancy further increases the risk of fetal complications, including macrosomia, neonatal hypoglycemia, hyperbilirubinemia, hypocalcemia, and polycythemia.

The physiological insulin resistance of pregnancy exacerbates glucose intolerance in diabetic mothers, leading to maternal hyperglycemia, which transfers to the fetus via the placenta, inducing fetal hyperglycemia. This, in turn, induces β cell hyperplasia in the fetal pancreas, resulting in chronic hyperglycemic hyperinsulinemia, a key mechanism underlying complications such as macrosomia. Moreover, pre-existing maternal diabetic complications, such as diabetic retinopathy or nephropathy, may worsen during pregnancy [8].

To prevent perinatal complications in both the mother and the fetus, planning pregnancy is important. Glycemic control should be optimized before the onset of the “absolute sensitive period,” and any pre-existing diabetic complications should be managed in advance by specialists to ensure that maternal health is suitable for pregnancy.

### 3.2. Overt Diabetes in Pregnancy

Overt diabetes in pregnancy, a category distinguished from gestational diabetes since 2010, includes (1) diabetes that was undiagnosed prior to pregnancy and (2) diabetes resulting from inadequate insulin secretion in response to pregnancy-induced insulin resistance. These women require more intensive management than those with gestational diabetes due to the higher likelihood of requiring insulin therapy, developing preeclampsia, and delivering large-for-gestational-age (LGA) infants. Additionally, women with overt diabetes tend to have a higher pre-pregnancy body mass index (BMI) and elevated HbA1c levels in early pregnancy, which correlate with increased risks of congenital anomalies relative to those with pregestational diabetes [9]. While strict blood glucose and weight management can mitigate some perinatal complications, including LGA, it is difficult to reduce congenital anomalies by treatment initiated after conception. Therefore, in cases of overt diabetes in pregnancy, it is important not only to maintain rigorous glycemic control during pregnancy but also to promote preconception screening among women of reproductive age.

### 3.3. Gestational Diabetes

Gestational diabetes refers to dysglycemia that is identified or develops during pregnancy in women with no prior diagnosis of diabetes. During pregnancy, human placental lactogen (also known as human chorionic somatomammotropin) is secreted by the placenta, which induces physiological insulin resistance. Typically, the maternal pancreatic islet β cells compensate by increasing insulin secretion, maintaining glucose homeostasis. However, when this compensatory mechanism is inadequate, gestational diabetes develops.

The accurate diagnosis of gestational diabetes is critical for two main reasons: (1) to prevent the perinatal complications related to maternal hyperglycemia and (2) to identify women at elevated risk of developing type 2 diabetes in the future. In 2010, the International Association of Diabetes and Pregnancy Study Groups (IADPSG) introduced diagnostic criteria based on the findings from the Hyperglycemia and Adverse Pregnancy Outcome (HAPO) study, which demonstrated that even mild hyperglycemia during pregnancy is associated with various perinatal complications. A diagnosis of gestational diabetes is made using a 75 g oral glucose tolerance test (OGTT), with gestational diabetes confirmed if any of the following thresholds are met or exceeded: fasting blood glucose ≥ 92 mg/dL, 1 h blood glucose ≥ 180 mg/dL, or 2 h blood glucose ≥ 153 mg/dL.

Gestational diabetes affects approximately 14% of all pregnancies. Compared to women with normal glucose tolerance, those with gestational diabetes have higher incidences of preeclampsia, gestational hypertension, macrosomia, neonatal respiratory distress, hypoglycemia, hyperbilirubinemia, and birth trauma [10,11]. The identified risk factors for gestational diabetes include advanced maternal age (≥35 years), pre-pregnancy obesity (BMI ≥ 25 kg/m^2^), a previous history of gestational diabetes, prior delivery of a macrosomic infant, a family history of diabetes, polycystic ovary syndrome, and ethnicity [12].

## 4. Classification of Type 1 Diabetes

Type 1 diabetes is etiologically classified into two main forms: (A) autoimmune type 1 diabetes, in which pancreatic islet β cells are destroyed through autoimmune mechanisms, and (B) idiopathic type 1 diabetes, characterized by the depletion of endogenous insulin secretion without detectable autoimmune activity. Autoimmune involvement is identified by the presence of anti-islet autoantibodies in the peripheral blood, including autoantibodies against glutamic acid decarboxylase (GADA), insulinoma-associated antigen-2 (IA-2A), zinc transporter 8 (ZnT8A), insulin autoantibodies (IAA), and islet cell antibodies (ICA). Clinically, type 1 diabetes is further classified based on the speed of β cell destruction into three subtypes: fulminant type 1 diabetes, acute-onset type 1 diabetes, and slowly progressive type 1 diabetes (slowly progressive insulin-dependent diabetes mellitus; SPIDDM), also referred to as latent autoimmune diabetes in adults (LADA) (Table 1). Fulminant type 1 diabetes progresses over “days”, acute-onset type 1 diabetes over “weeks”, and SPIDDM/LADA over “months to years” [13].

### 4.1. Acute-Onset Type 1 Diabetes

Acute-onset type 1 diabetes is the classic form of the disease and is typically triggered by a combination of genetic susceptibility, particularly involving human leukocyte antigen (HLA) genes, and environmental factors such as viral infections. In healthy individuals, autoreactive T cells are eliminated through a process known as negative selection in the thymus. Even if these cells evade thymic deletion and enter the peripheral circulation, they are typically restrained by regulatory T cells (Tregs), which suppress autoimmune responses. However, in genetically predisposed individuals, this immune balance may be disrupted, leading to a targeted autoimmune attack on the pancreatic islet β cells. Acute-onset type 1 diabetes often manifests during infancy, school age, or adolescence. However, a recent database study indicates that the incidence of type 1 diabetes in adulthood is comparable to that in childhood [14], indicating that type 1 diabetes can occur at any age.

As outlined in Table 1, this subtype is defined by the onset of diabetic ketosis or ketoacidosis within three months of the initial hyperglycemic symptoms, followed by a persistent requirement for insulin therapy. Although residual endogenous insulin secretion may still be detectable at diagnosis, the insulin secretory capacity typically declines, progressing to complete insulin dependence. For diagnostic evaluation, it is important that the endogenous insulin secretion is evaluated by C-peptide levels and anti-islet autoantibodies, with approximately 95% of cases classified as autoimmune type 1 diabetes [13].

### 4.2. Slowly Progressive Type 1 Diabetes (SPIDDM)

SPIDDM is characterized by a gradual decline in insulin secretion. Patients do not present with diabetic ketosis or ketoacidosis at diagnosis and initially do not require insulin therapy. However, insulin dependence develops progressively, typically over three to six months or longer. SPIDDIM is frequently misdiagnosed as type 2 diabetes at onset. However, in certain cases, such as soft drink ketosis or coexisting autoimmune thyroid disease, patients may initially present with ketosis.

This subtype predominantly manifests in adulthood, although pediatric cases may be identified during routine urine glucose screening at school or during health checkups. Diagnosis requires the presence of at least one positive anti-islet autoantibody, such as ICA, IAA, GADA, IA-2A, or ZnT8A. Among these, GADA are the most prevalent, and single positivity for GADA accounts for more than one-third of cases. Therefore, GADA testing is recommended as the initial screening measure in cases suspected of SPIDDM. Combined antibody testing increases the diagnostic sensitivity, and SPIDDM accounts for approximately 12% of insulin-naïve diabetic patients [13].

Endogenous insulin secretion typically declines over time in SPIDDM, but some patients remain insulin-independent for a decade or longer. Based on their insulin status at the most recent follow-up, patients are categorized as “SPIDDM (definite)” if they are insulin-dependent (fasting C-peptide < 0.6 ng/mL) or “SPIDDM (probable)” if they remain insulin-independent (Table 1) [15].

Identifying patients at high risk for future insulin dependence is crucial during the SPIDDM (probable) stage. According to a report by Kawasaki et al., positivity for multiple anti-islet autoantibodies is a strong predictor of progression to insulin dependence. Additional risk factors include a lower BMI, younger age at onset, a shorter interval between symptom onset and diagnosis, and lower C-peptide levels at diagnosis [15].

### 4.3. Fulminant Type 1 Diabetes

Fulminant type 1 diabetes is a subtype in which the pancreatic islet β cells are almost completely destroyed within approximately one week, resulting in sudden and marked hyperglycemia. This subtype accounts for approximately 20% of ketosis-onset type 1 diabetes cases in Japan. While this subtype is rare in children, it is more common in East Asian than in Western Caucasian populations.

The duration of hyperglycemic symptoms is typically very short, with a mean (±SD) of 4.4 ± 3.1 days. Despite extremely high mean blood glucose levels at onset (800 ± 360 mg/dL), the mean HbA1c level is near normal (6.4% ± 0.9%), reflecting the rapid disease progression. Over 95% of affected individuals exhibit elevated pancreatic exocrine enzymes, leading to frequent misdiagnosis as acute pancreatitis [16]. Viral infection is thought to be closely linked to the onset of fulminant type 1 diabetes. It may also occur concurrently with drug-induced hypersensitivity syndrome or drug reaction with eosinophilia and systemic symptoms (DRESS), both of which involve viral reactivation, such as that of human herpes virus-6 [17]. In addition, fulminant type 1 diabetes has been reported as an immune-related adverse event following treatment with immune checkpoint inhibitors [18].

Although anti-islet autoantibodies are typically negative in fulminant type 1 diabetes, occasional positive cases have been reported. Therefore, anti-islet autoantibodies are not included in the diagnostic criteria for fulminant type 1 diabetes.

## 5. Genetic Factors

### 5.1. Genetic Factors of Gestational Diabetes

As discussed above, gestational diabetes arises from a combination of environmental and genetic risk factors. While the environmental factors, such as age, obesity, and lifestyle, are well-established, accumulating evidence suggests that genetic predisposition also plays a significant role in the development of gestational diabetes. Although the impaired glucose tolerance associated with gestational diabetes typically resolves following delivery, women with a history of gestational diabetes are at a substantially increased risk of developing type 2 diabetes postpartum than women who have had normal pregnancies. Longitudinal studies indicate that 35–60% of women with prior gestational diabetes develop type 2 diabetes within 10 to 20 years after childbirth. These data suggest that genetic background is involved in impaired glucose tolerance in gestational diabetes.

Genetic association studies investigating susceptibility loci for type 2 diabetes have also identified shared genetic markers in women with gestational diabetes. In European Caucasian populations, polymorphisms in genes such as CDK5 regulatory subunit-associated protein 1-like 1 (*CDKAL1*), cyclin-dependent kinase inhibitor 2A/2B (*CDKN2A/2B*), transcription factor 7-like 2 (*TCF7L2*), and melatonin receptor 1B (*MTNR1B*) have been implicated in the development of gestational diabetes [19]. In Japanese populations, the gestational-diabetes-related loci include adiponectin (*ADIPOQ*), *CDKN2A/2B*, signal sequence receptor subunit 1-Ras-responsive element binding protein 1 (*SSR1-RREB1*), and MyoD family inhibitor domain-containing 2 (*MDFIC2*) [20,21].

### 5.2. Genetic Factors of Type 1 Diabetes

The incidence of type 1 diabetes varies significantly across ethnic groups, with the highest rates observed in Northern European populations and the lowest in Asian populations, including Japan. This disparity strongly suggests a genetic contribution to disease susceptibility.

Among the various genetic determinants, the HLA region, especially the HLA class II DR and DQ genes, has been shown to confer the greatest risk. In Western Caucasians, the haplotypes HLA-*DRB1*03:01-DQB1*02:01* and *DRB1*04:01-DQB1*03:02* are strongly associated with susceptibility to type 1 diabetes. However, these haplotypes are extremely rare in the Japanese population. In contrast, the major risk haplotypes in Japanese patients include HLA-*DRB1*04:05–DQB1*04:01*, *DRB1*08:02–DQB1*03:02* and *DRB1*09:01–DQB1*03:03*. Protective haplotypes also differ by population: *DRB1*15:01–DQB1*06:02* and *DRB1*15:02–DQB1*06:01*, both under the HLA–DR2 category, are negatively associated with type 1 diabetes in Japanese individuals. Notably, *DRB1*15:01–DQB1*06:02* is considered an almost universal protective haplotype.

The association between class II HLA haplotypes and disease susceptibility also varies by subtype of type 1 diabetes. For both the acute-onset and slowly progressive forms, susceptibility is linked to *DRB1*04:05–DQB1*04:01*, *DRB1*08:02–DQB1*03:02*, and *DRB1*09:01–DQB1*03:03*. In contrast, fulminant type 1 diabetes displays a distinct genetic profile. Only *DRB1*04:05–DQB1*04:01* has been associated with increased risk, while the protective *DRB1*15:01–DQB1*06:02* haplotype does not confer resistance in this subtype [22].

Beyond the HLA genes, more than 100 non-HLA loci for type 1 diabetes have been identified through linkage studies, candidate gene association studies, and genome-wide association studies (GWAS). Notably, the variable number of tandem repeats in the insulin gene promoter (*INS-VNTR*), as well as variants in cytotoxic T lymphocyte antigen-4 (*CTLA-4*), interleukin 2 receptor subunit alpha (*IL2RA*), and protein tyrosine phosphatase non-receptor type 22 (*PTPN22*), have been implicated in Western populations [23]. Many of these loci also confer risk in Japanese individuals; however, there are significant differences in the associated phenotypes and risk allele frequencies across populations, highlighting the need for population-specific genetic investigations [22].

## 6. Predictive Markers for Type 1 Diabetes

Type 1 diabetes is characterized by an autoimmune attack on the pancreatic islet β cells, resulting in their progressive destruction by cytotoxic T cells. As this autoimmune process advances, several types of anti-islet autoantibodies appear in the peripheral blood, serving as surrogate biomarkers for the destruction of pancreatic islet β cells. These include IAA, ICA, GADA, IA-2A, and ZnT8A.

In acute-onset type 1 diabetes, the number of positive autoantibodies is a more significant predictor of disease onset than the specificity of any single autoantibody. Individuals who are positive for two or more autoantibodies are at a substantially higher risk than those who are positive for a single autoantibody. Longitudinal studies have shown that the five-year risk of developing type 1 diabetes is approximately 44% among individuals with normal glucose tolerance (Stage 1) who are positive for multiple autoantibodies. This risk increases dramatically in individuals with impaired glucose tolerance (Stage 2), with a two-year risk of 60% and a five-year risk of 75%.

In contrast, for patients with SPIDDM who have already been diagnosed with diabetes, anti-islet autoantibodies are used as predictive markers for progression to an insulin-deficient state. Several factors have been identified as predictors of this transition, including high titers of GADA, which are the most prevalent autoantibodies in SPIDDM, positivity for multiple autoantibodies, especially combinations including GADA, and high-affinity GADA, which are associated with more rapid β cell deterioration [13]. These markers are valuable not only for identifying individuals at risk but also for stratifying patients by their likelihood of disease progression, thereby supporting earlier and more personalized interventions.

## 7. Pregnancy and Immunity

Pregnancy presents a unique immunological challenge, as the maternal immune system must tolerate the “non-self” fetus while maintaining a defense against external pathogens. This tolerance is mediated in part by maternal hormones such as estrogen or progesterone, which modulate immune responses to prevent fetal rejection [24,25,26].

Traditionally, immune regulation during pregnancy has been described in terms of the balance between T-helper 1 (Th1) and T-helper 2 (Th2) cells. Th1 cells activate cytotoxic T cells that target endogenous “non-self” cells, while Th2 cells promote antibody production against exogenous pathogens. During pregnancy, the maternal immune system shifts toward Th2 dominance, fostering immunological tolerance to the fetus. As estrogen suppresses Th1 cell activity while progesterone enhances Th2 cell activation, these hormones have been credited with regulating the Th1/Th2 balance through interactions with dendritic cells [24,25].

More recent research, however, has focused on the roles of Tregs and T-helper 17 (Th17) cells in early-pregnancy-related immune tolerance [27,28]. Tregs suppress the autoimmune response, while Th17 cells promote inflammatory and autoimmune reactions. The balance between these cell types, favoring Tregs, is essential for successful embryo implantation and the maintenance of early pregnancy (Figure 1). Notably, Tregs and Th17 cells are also critically involved in the pathogenesis of type 1 diabetes. This condition is characterized by the immune-mediated destruction of pancreatic islet β cells, and dysregulation in the Th17/Treg balance plays a key role in disease development [29,30,31]. Specifically, increased Th17 cell proliferation and impaired Tregs function have been reported as hallmark immune disturbances in patients with type 1 diabetes [32,33].

## 8. Pregnancy-Associated Fulminant Type 1 Diabetes

Type 1 diabetes may first manifest during pregnancy or shortly after delivery in individuals with no prior history of diabetes (Figure 2).

In a nationwide survey involving 286 Japanese women of childbearing age (13–49 years) with type 1 diabetes, approximately 4% of cases fell into pregnancy-associated type 1 diabetes [34]. Notably, 75% of these cases were classified as fulminant type 1 diabetes [35]. This proportion is significantly higher than that observed in non-pregnancy-associated type 1 diabetes (9.1%), with an odds ratio of 29.9 (*p* = 5.2 × 10^−12^) (Figure 3).

Compared to acute-onset type 1 diabetes [36], pregnancy-associated fulminant type 1 diabetes is characterized by a shorter duration of hyperglycemic symptoms before diagnosis, relatively low HbA1c levels despite severe hyperglycemia, and low urinary C-peptide concentrations at onset [37]. However, when comparing pregnancy-associated and non-pregnancy-associated fulminant type 1 diabetes, no significant differences were found in the clinical parameters at diagnosis, including age at onset, BMI, family history of diabetes, duration of hyperglycemic symptoms, frequency of flu-like symptoms, blood glucose levels, frequency of GADA, or C-peptide levels [38].

However, pregnancy-associated fulminant type 1 diabetes is distinguished by significantly lower arterial blood pH, elevated serum amylase levels, and significantly higher prevalence of the HLA-*DRB1*09:01-DQB1*03:03* haplotype compared to non-pregnancy-associated fulminant type 1 diabetes [38]. This condition most commonly develops between the third trimester and two weeks postpartum, with fetal mortality reported in approximately 67% of cases [38].

Given the high risks posed to both mother and fetus, fulminant type 1 diabetes should be strongly suspected in pregnant or postpartum women presenting with marked hyperglycemia (≥288 mg/dL) and relatively low HbA1c levels (<8.7%) despite no prior history of diabetes. In such cases, the prompt initiation of insulin therapy, in coordination with internal medicine and obstetrics teams, is essential to improving maternal and fetal outcomes.

## 9. Etiology of Pregnancy-Associated Fulminant Type 1 Diabetes

The etiology of fulminant type 1 diabetes remains incompletely understood; however, more than 70% of affected individuals report flu-like symptoms (e.g., fever, upper respiratory tract symptoms) preceding the onset of hyperglycemia, suggesting that viral infections are a major triggering factor. Furthermore, a patient’s high prevalence of the class II HLA haplotype (HLA-*DRB1*04:05-DQB1*04:01*) indicates a strong genetic predisposition [16].

Further evidence supporting the involvement of innate immunity includes the elevated expression of pattern recognition receptors such as melanoma differentiation–associated gene-5 (MDA5), retinoic acid–inducible gene I (RIG-I), toll-like receptor 3 (TLR3), and TLR4 in the pancreatic islets of patients with fulminant type 1 diabetes [39]. These findings suggest that both genetic susceptibility and an exaggerated innate antiviral response contribute to disease pathogenesis.

Additionally, patients with fulminant type 1 diabetes exhibit significantly lower serum levels of soluble CTLA-4, a key negative regulator of immune responses, compared to those with acute-onset type 1 diabetes [40]. Reduced CTLA-4 expression on CD4^+^ helper T cells has also been observed [41], implying impaired immune tolerance and dysregulated antiviral responses, which may accelerate the pancreatic islet β cell destruction.

In a physiological pregnancy, Somerset and coworkers reported that circulating Tregs, essential for maintaining maternal–fetal immune tolerance, increase during early gestation, peak in the second trimester, and decline in the postpartum period [42]. Notably, pregnancy-associated fulminant type 1 diabetes often develops in the third trimester or shortly after delivery [43], coinciding with the decline in circulating Tregs. This temporal association supports the hypothesis that viral infection during this immunologically vulnerable period may trigger an exaggerated immune response, leading to pancreatic islet β cell destruction and the onset of fulminant type 1 diabetes (Figure 4).

## 10. Etiology of Autoimmune Gestational Diabetes

Women diagnosed with gestational diabetes who test positive for anti-islet autoantibodies, such as GADA, are considered to have “autoimmune gestational diabetes” [44,45]. These individuals are at a high risk of progressing to SPIDDM/LADA postpartum (Figure 2). SPIDDM/LADA is a subtype of type 1 diabetes characterized by the slow autoimmune destruction of pancreatic islet β cells, typically without requiring insulin at the time of diagnosis. As a result, these patients are often initially misdiagnosed with type 2 diabetes and only correctly classified as SPIDDM/LADA after the detection of anti-islet autoantibodies.

Multicenter studies have reported that 4–14% of patients initially diagnosed with type 2 diabetes actually have SPIDDM/LADA [13,46,47]. Furthermore, recent meta-analyses indicate that the risk of progression from gestational diabetes to type 2 diabetes is 6 to 10 times higher in women with gestational diabetes than in the general obstetric population [48,49]. It is presumed that a subset of these cases represents undiagnosed autoimmune gestational diabetes.

Previous studies reported that cell-mediated immunity plays a critical role in the etiology of SPIDDM/LADA [50], and anti-islet autoantibodies are considered immune markers generated in response to autoantigens released into the blood during immune-mediated pancreatic islet β cell destruction. However, the exact mechanisms underlying autoantibody production in humans remain largely unclear.

One hypothesis regarding the etiology of autoimmune gestational diabetes is that pregnancy itself alters the autoimmune response, triggering a de novo autoimmune response against pancreatic islet β cell antigens. However, a more widely accepted explanation is that islet β cell autoimmunity exists prior to pregnancy but remains latent. The increased insulin resistance associated with pregnancy then serves as a physiological stressor, unmasking glucose intolerance and leading to the clinical diagnosis of gestational diabetes. From this perspective, autoimmune gestational diabetes represents a preclinical stage in the natural progression toward type 1 diabetes.

## 11. Autoimmune Gestational Diabetes as a Predictor of Type 1 Diabetes Development

The majority of autoimmune gestational diabetes cases progress to SPIDDM/LADA, making it difficult to distinguish from non-autoimmune gestational diabetes based solely on its clinical presentation.

In our retrospective analysis of a patient with Graves’ disease who later developed type 1 diabetes following two episodes of gestational diabetes, multiple anti-islet autoantibodies were already detectable before the initial diagnosis of gestational diabetes [51]. This finding highlights the importance of testing for anti-islet autoantibodies at the time of the gestational diabetes diagnosis to facilitate the early detection of latent islet autoimmunity.

In Western Caucasian populations, numerous studies have shown that 5–10% of women with gestational diabetes test positive for at least one anti-islet autoantibody when a combination of GADA, IA-2A, and ZnT8A is measured (Table 2) [52,53,54,55,56,57]. In contrast, data from Japanese populations remain limited. The reported GADA positivity rates in Japanese women are 0.3–3.5% in gestational diabetes and approximately 0.8% in overt diabetes during pregnancy [58,59].

A 23-year follow-up study of 391 women with gestational diabetes, including 58 classified as autoimmune gestational diabetes, demonstrated a significantly increased postpartum risk of developing type 1 diabetes in correlation with the number of positive anti-islet autoantibodies. Remarkably, all women who tested positive for three autoantibodies out of ICA, GADA, IA-2A, and ZnT8A progressed to type 1 diabetes within seven years [60].

Additional independent risk factors for progression to type 1 diabetes included ICA and GADA positivity, a diagnosis of gestational diabetes before age 30, and a requirement for insulin therapy during pregnancy.

Accordingly, in young women diagnosed with gestational diabetes, particularly those requiring insulin therapy during pregnancy or with coexisting autoimmune conditions such as autoimmune thyroid disease, routine testing for anti-islet autoantibodies, particularly GADA, should be considered (Figure 5). If positive, close monitoring of blood glucose levels, HbA1c, and endogenous insulin secretion is essential for predicting and potentially delaying the onset of type 1 diabetes.

## 12. Conclusions

At present, there are no established predictive markers for fulminant type 1 diabetes prior to clinical onset. As a result, early detection and timely intervention are critical to safeguarding the health of both the mother and the fetus. Future research efforts aimed at identifying reliable predictive biomarkers for fulminant type 1 diabetes may facilitate early diagnosis and enable the development of preventative strategies. However, due to the ethnic or geographic characteristics of fulminant type 1 diabetes, there are also limitations to the transferability to populations outside of East Asia, and some conclusions may not be generalizable.

Autoimmune gestational diabetes is not uncommon, affecting 5–10% of women with gestational diabetes. Although its clinical presentation is largely indistinguishable from that of non-autoimmune gestational diabetes, the progressive impairment of pancreatic islet β cell function leads to a gradual decline in endogenous insulin secretion. This results in unpredictable hyperglycemia and hypoglycemia, leading to a significant deterioration in quality of life. Therefore, early diagnosis by measuring anti-islet autoantibodies and initiating insulin treatment at the appropriate time are crucial to maintaining optimal glycemic control and preventing long-term diabetic complications.

We anticipate that future large-scale studies, particularly those incorporating genetic analyses and the search for predictive immune markers, will provide new insights into the pathogenesis of autoimmune gestational diabetes. These advances may enable earlier detection, individualized intervention, and improved clinical outcomes for patients at risk of progressing to type 1 diabetes.

## Figures and Tables

**Figure 1 ijms-26-06499-f001:**
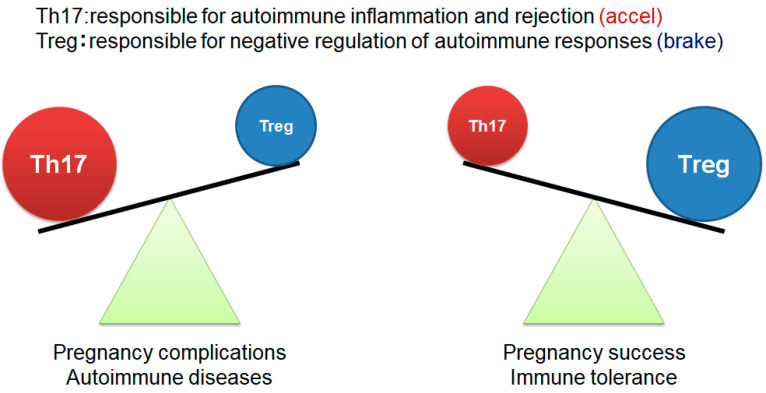
Th17/Treg balance in the establishment and maintenance of pregnancy The figure was adapted from Ref. [27].

**Figure 2 ijms-26-06499-f002:**
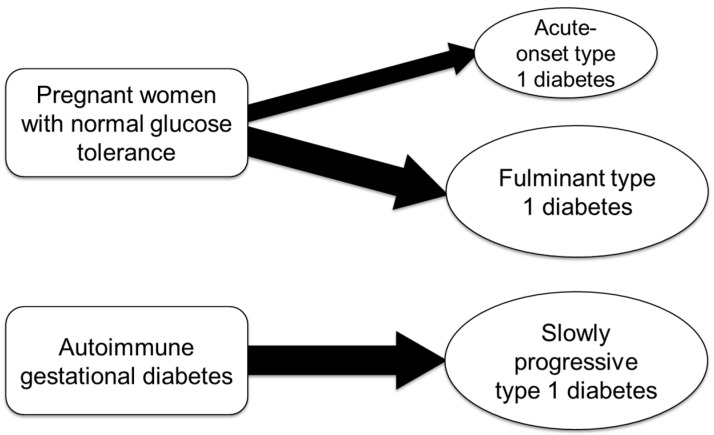
Subtypes of pregnancy-associated type 1 diabetes.

**Figure 3 ijms-26-06499-f003:**
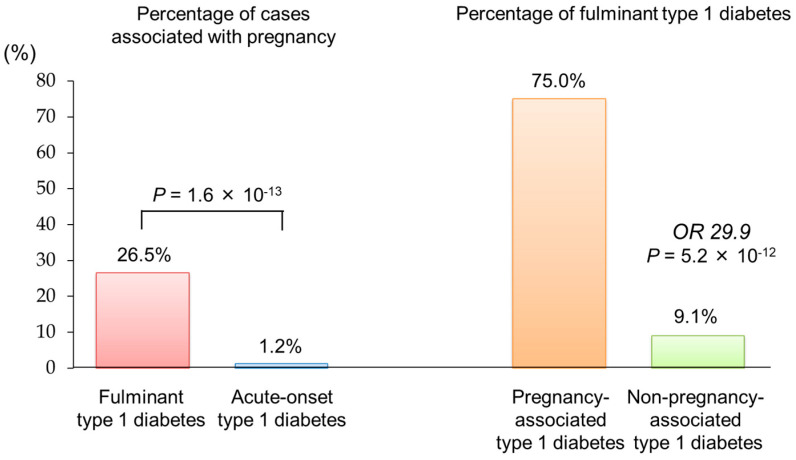
Association between fulminant type 1 diabetes and pregnancy The figure was adapted from Ref. [34].

**Figure 4 ijms-26-06499-f004:**
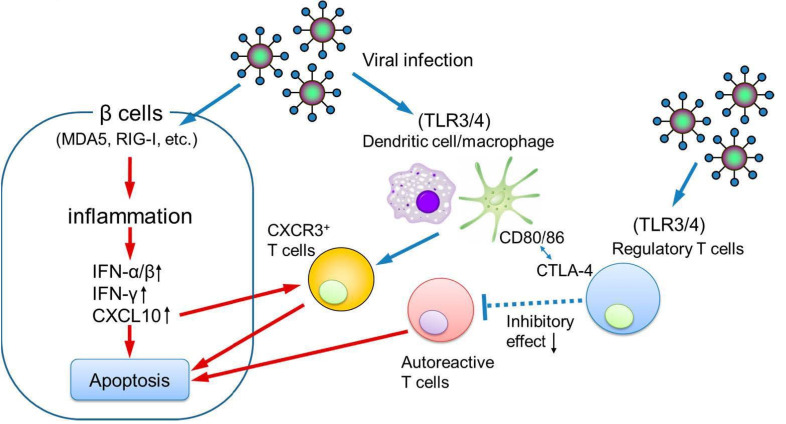
The possible pathogenesis of pregnancy-associated fulminant type 1 diabetes. IFN, interferon; CXCL10, C-X-C motif chemokine ligand 10; CXCR3, C-X-C motif chemokine receptor 3.

**Figure 5 ijms-26-06499-f005:**
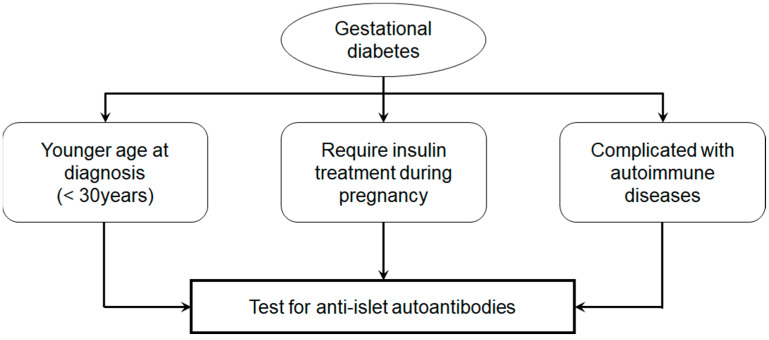
Prediction of type 1 diabetes in gestational diabetes.

**Table 1 ijms-26-06499-t001:** Diagnostic criteria for acute-onset, slowly progressive, and fulminant type 1 diabetes.

Acute-Onset Type 1 Diabetes	SPIDDM	Fulminant Type 1 Diabetes
Clinical symptoms and the need for insulin treatment		
1. Occurrence of diabetic ketosis or ketoacidosis around <3 months after the onset of hyperglycemic symptoms 2. Need for continuous insulin therapy after the diagnosis of diabetes mellitus. A temporary honeymoon period may occur.	1. The absence of ketosis or ketoacidosis at the diagnosis of diabetes and the lack of need for insulin treatment to correct hyperglycemia immediately after diagnosis in principle	1. Occurrence of diabetic ketosis or ketoacidosis soon (around 7 days) after the onset of hyperglycemic symptoms
Blood glucose‧HbA1c‧anti-islet autoantibodies (Note 1)		
3. Positive test result for anti-islet autoantibodies	2. Positive test result for anti-islet autoantibodies at some time point during the disease course	2. Plasma glucose level ≥ 288 mg/dL and HbA1c level < 8.7% at first visit (Note 2)
Endogenous insulin secretion		
4. Presence of endogenous insulin deficiency (fasting serum C-peptide < 0.6 ng/mL) without verifiable anti-islet autoantibodies	3. Gradual decline in insulin secretion over time, no requirement for insulin treatment for ≥3 months (typically ≥ 6 months) after diagnosis of diabetes, and severe endogenous insulin deficiency (fasting serum C-peptide < 0.6 ng/mL) at last observed time point	3. Urinary C-peptide excretion < 10 µg/day or fasting serum C-peptide level < 0.3 ng/mL and <0.5 ng/mL after intravenous glucagon (or after meal) load at onset
Diagnosis		
“Acute-onset type 1 diabetes mellitus (autoimmune)”: fulfilled criteria 1, 2, and 3 “Acute-onset type 1 diabetes mellitus”: fulfilled criteria 1, 2, and 4	“SPIDDM (definite)”: fulfilled criteria 1, 2, and 3 “SPIDDM (probable)”: fulfilled criteria 1 and 2 only, but not 3	“Fulminant type 1 diabetes”: fulfilled criteria 1, 2, and 3

(Note 1) These include islet cell antibodies (ICA), autoantibodies to glutamic acid decarboxylase (GADA), insulinoma-associated antigen-2 (IA-2A), zinc transporter 8 (ZnT8A), or insulin autoantibodies (IAA). IAA should be evaluated before or shortly after insulin therapy is initiated. (Note 2) This value is not applicable to patients with previously diagnosed glucose intolerance.

**Table 2 ijms-26-06499-t002:** Prevalence of anti-islet autoantibodies in women with gestational diabetes.

Country	Number of Cases	GADA	IA-2A	ZnT8A	Reference
Germany	437	9.5%	6.2%		[52]
Sweden	385	6.0%	1.8%		[53]
Sweden	193	6.2%	2.6%	2.6%	[54]
Sweden	281	11.4%	3.6%		[55]
Denmark	407	5.4%			[56]
Australia	302	2.3%	2.0%	4.8%	[57]
Total	2005	6.8%	3.4%	3.9%	

GADA, glutamic acid decarboxylase autoantibody; IA-2A, insulinoma-associated antigen-2 autoantibody; ZnT8A, zinc transporter 8 autoantibody.

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
