# Peer review of "Fulminant and Slowly Progressive Type 1 Diabetes Associated with Pregnancy"

_ijms, 2025, doi:10.3390/ijms26136499_

Round 1
Reviewer 1 Report
Comments and Suggestions for Authors
The manuscript deals with a relevant and little-addressed topic in the international literature, namely the association between type 1 diabetes and pregnancy, with a focus on fulminant and slowly progressing forms. The treatment is well structured, up-to-date and supported by a relevant bibliography. The sections on pathophysiology, genetics and immunological aspects are well integrated and are of great use to the clinical and research reader. However, to further improve the work, the criteria used for the selection of literature should be clearly introduced. In particular, the author should indicate whether a systematic or narrative search was conducted, specifying the databases consulted, the keywords used, any time or language restrictions, and the inclusion and exclusion criteria of the studies cited. This aspect is crucial to ensure the transparency of the selection process and lend more methodological soundness to the review. The title could be more specific by explicitly referring to "fulminant" and "slowly progressive" type 1 diabetes forms associated with pregnancy. Furthermore, although most of the evidence comes from studies conducted in Japanese populations, it would be useful to briefly discuss the transferability of the findings to other ethnic or geographical contexts, or to acknowledge within the limits of the work that some conclusions may not be generalisable. Finally, the conclusion could be expanded to emphasise the concrete clinical implications, both in terms of early diagnosis and management. Overall, the work represents a valuable and well-documented contribution, but can be further strengthened through greater methodological transparency and a broader reflection on the applicability of the results.
Author Response
I thank you for your evaluation of our manuscript. I have revised the manuscript according to your comments.
Comments and Suggestions for Authors
The manuscript deals with a relevant and little-addressed topic in the international literature, namely the association between type 1 diabetes and pregnancy, with a focus on fulminant and slowly progressing forms. The treatment is well structured, up-to-date and supported by a relevant bibliography. The sections on pathophysiology, genetics and immunological aspects are well integrated and are of great use to the clinical and research reader.
However, to further improve the work, the criteria used for the selection of literature should be clearly introduced. In particular, the author should indicate whether a systematic or narrative search was conducted, specifying the databases consulted, the keywords used, any time or language restrictions, and the inclusion and exclusion criteria of the studies cited. This aspect is crucial to ensure the transparency of the selection process and lend more methodological soundness to the review.
Response: The type of this review paper is a “State-of-the-Art review” article, but not systematic review or narrative review. Therefore, I did not go through the literature selection process you mentioned.
The title could be more specific by explicitly referring to "fulminant" and "slowly progressive" type 1 diabetes forms associated with pregnancy.
Response: According to your suggestion, I changed the title to “Fulminant and slowly-progressive type 1 diabetes associated with pregnancy”.
Furthermore, although most of the evidence comes from studies conducted in Japanese populations, it would be useful to briefly discuss the transferability of the findings to other ethnic or geographical contexts, or to acknowledge within the limits of the work that some conclusions may not be generalizable.
Response: Based on your comment, I added the following sentence in the “Conclusions” section.
“However, due to the ethnic or geographic characteristics of fulminant type 1 diabetes, there are also limitations to the transferability to populations outside of East Asia, and some conclusions may not be generalizable.”
Finally, the conclusion could be expanded to emphasize the concrete clinical implications, both in terms of early diagnosis and management.
Response: According to your comment, I modified the conclusion as follows,
“This results in unpredictable hyperglycemia and hypoglycemia, leading to a significant deterioration in their quality of life. Therefore, early diagnosis by measuring anti-islet autoantibodies and initiating insulin treatment at the appropriate time are crucial to maintaining optimal glycemic control and preventing long-term diabetic complications.”
Reviewer 2 Report
Comments and Suggestions for Authors
Recommendation:
- High percentage match, please reduce it.
- There could be more constructive figures, the one presented do not illustrate pathophysiological mechanisms enough.
- Also if the figures illustrate findings from different studies, you should put the reference in the caption, since is not your original data!
Author Response
I thank you for your evaluation of our manuscript. I have revised the manuscript according to your comments.
Comments and Suggestions for Authors
- High percentage match, please reduce it.
Response: I extensively scanned my manuscript using a Plagiarism Checker (https://www.duplichecker.com/) and have made efforts to reduce the match rate. The revised manuscript was marked up using the “Track Changes” function.
- There could be more constructive figures, the one presented do not illustrate pathophysiological mechanisms enough.
Response: According to your comment, I created and replaced Figure 4 that illustrates the possible mechanisms of pregnancy-associated fulminant type 1 diabetes.
- Also if the figures illustrate findings from different studies, you should put the reference in the caption, since is not your original data!
Response: Thank you for your comment. Where applicable, I have added the references to the figure captions.
Round 2
Reviewer 1 Report
Comments and Suggestions for Authors
The manuscript covers a clinically relevant and underexplored topic with solid integration of genetic, immunological, and clinical aspects. However, the critical concern remains unresolved: there is still no explicit section describing the literature search strategy (databases, keywords, inclusion/exclusion criteria). Even if classified as a “state-of-the-art review,” this should be clearly stated in the methods or introduction to enhance transparency and reproducibility. I encourage the authors to add a brief methodological note clarifying this point.
Author Response
I thank you for your re-evaluation of our manuscript. I have revised the manuscript according to your comment.
Comments and Suggestions for Authors
The manuscript covers a clinically relevant and underexplored topic with solid integration of genetic, immunological, and clinical aspects.
However, the critical concern remains unresolved: there is still no explicit section describing the literature search strategy (databases, keywords, inclusion/exclusion criteria). Even if classified as a “state-of-the-art review,” this should be clearly stated in the methods or introduction to enhance transparency and reproducibility. I encourage the authors to add a brief methodological note clarifying this point.
Response: I added the “Literature search strategy” section in the re-revised manuscript.
Round 3
Reviewer 1 Report
Comments and Suggestions for Authors
All my concerns have been addressed